# Genotypic analysis of the female BPH/5 mouse, a model of superimposed preeclampsia

Jenny L. Sones[1]*, Christina C. Yarborough[2], Valerie O'Besso[3], Alexander Lemenze[4,5], Nataki C. Douglas[5,6]

1 Veterinary Clinical Sciences, Louisiana State University School of Veterinary Medicine, Baton Rouge, LA, United States of America, 2 School of Graduate Studies, Rutgers Biomedical and Health Sciences, Newark, NJ, United States of America, 3 Rutgers New Jersey Medical School, Newark, NJ, United States of America, 4 Department of Pathology, Immunology, and Laboratory Medicine, Rutgers New Jersey Medical School, Newark, NJ, United States of America, 5 Center for Immunity and Inflammation, Rutgers Biomedical and Health Sciences, Newark, NJ, United States of America, 6 Department of Obstetrics, Gynecology and Reproductive Health, Rutgers New Jersey Medical School, Newark, NJ, United States of America

* jsones@lsu.edu

**Data Availability Statement:** All raw files are available from the short reads archives (PRJNA701765).

## Abstract

Animal models that recapitulate human diseases and disorders are widely used to investigate etiology, diagnosis, and treatment of those conditions in people. Disorders during pregnancy are particularly difficult to explore as interventions in pregnant women are not easily performed. Therefore, models that allow for pre-conception investigations are advantageous for elucidating the mechanisms involved in adverse pregnancy outcomes that are responsible for both maternal and fetal morbidity, such as preeclampsia. The Blood Pressure High (BPH)/5 mouse model has been used extensively to study the pathogenesis of preeclampsia. The female BPH/5 mouse is obese with increased adiposity and borderline hypertension, both of which are exacerbated with pregnancy making it a model of superimposed preeclampsia. Thus, the BPH/5 model shares traits with a large majority of women with pre-existing conditions that predisposes them to preeclampsia. We sought to explore the genome of the BPH/5 female mouse and determine the genetic underpinnings that may contribute to preeclampsia-associated phenotypes in this model. Using a whole genome sequencing approach, we are the first to characterize the genetic mutations in BPH/5 female mice that make it unique from the closely related BPH/2 model and the normotensive background strain, C57Bl/6. We found the BPH/5 female mouse to be uniquely different from BPH/2 and C57Bl/6 mice with a genetically complex landscape. The majority of non-synonymous consequences within the coding region of BPH/5 females were missense mutations found most abundant on chromosome X when comparing BPH/5 and BPH/2, and on chromosome 8 when comparing BPH/5 to C57Bl/6. Genetic mutations in BPH/5 females largely belong to immune system-related processes, with overlap between BPH/5 and BPH/2 models. Further studies examining each gene mutation during pregnancy are warranted to determine key contributors to the BPH/5 preeclamptic-like phenotype and to identify genetic similarities to women that develop preeclampsia.

**Funding:** SOURCES OF FUNDING NIH COBRE P20 GM135002 (JLS); NIH R01 HL127013 (NCD).

**Competing interests:** The authors have declared that no competing interests exist.

## Introduction

Genetically hypertensive mice have long been utilized as translational models to investigate blood pressure regulation. The closely related strains Blood Pressure High (BPH)/2, Blood Pressure Normal (BPN)/3, and Blood Pressure Low (BPL)/1 mice, with different cardiovascular and metabolic profiles, are examples [1]. These strains were established after 23 generations of two-way selection for high and low systolic blood pressure. The base population for this selection scheme was derived from an 8-way cross between C57Bl/6 (C57), SJL, BDP, LP, RF, CBA, BALB/c, and 129 inbred mice [1] (**Fig 1**). Female BPH/2 mice are lean and hypertensive with systolic blood pressure and heart rates significantly higher than BPN/3 mice [2]. Female BPL/1 mice have hypotension compared to normotensive BPN/3 [3]. These strains have been used to investigate the mechanisms that lead to hypertension in humans [4–9]. From BPH/2, 13 sublines were created after >20 generations of brother-sister matings. One of these sublines was identified as BPH/5, which have lower blood pressures than BPH/2 mice, with female mice considered "borderline hypertensive" with mean arterial pressure ranging from 103 to 119 mmHg as measured by radiotelemetry [10,11].

More recent studies have shown that female BPH/5 mice have both metabolic and reproductive disorders [12]. By 8 weeks of age, they exhibit increased body mass with excessive subcutaneous and visceral white adipose tissue (WAT) depots [12,13], and they remain obese throughout adulthood. Further, adult female BPH/5 mice show hyperphagia and hyperleptinemia with signs of leptin resistance, suggesting metabolic syndrome and even insulin resistance. Despite being fecund, female BPH/5 mice have irregular estrous cycles, with low circulating estrogen, levels regardless of the stage of the estrous cycle, and inappropriate uterine response to sex steroid hormones [12].

In pregnancies generated from brother-sister matings, BPH/5 females spontaneously exhibit the cardinal features of preeclampsia (PE), maternal hypertension and proteinuria during the second half of gestation [10]. PE is a hypertensive disorder of human pregnancy, affecting up to 8% of women in the United States, that can cause maternal death (~76,000/year) and infant loss (~500,000/year) [14,15]. Superimposed PE occurs when pre-existing maternal hypertension is worsened with pregnancy and is accompanied by renal, liver, neurologic and/or hematologic dysfunction after 20 weeks gestation [15]. PE can have significant fetal co-morbidities, including fetal growth restriction, which carry long-term health consequences for the offspring into adult life [16]. PE is multifactorial and thus considered a complex syndrome. Furthermore, the presentation of PE takes several forms, early (<34 weeks of gestation) and late onset ($\geq$34 weeks of gestation). The etiology for early versus late onset is thought to involve two differing placental pathologies: shallow trophoblast invasion into maternal spiral arteries and placental villous overcrowding, respectively [17–19]. More severe fetal morbidity is associated with early onset PE, consistent with current understanding that placental pathology of early onset PE develops in the 1st trimester of pregnancy. Importantly, maternal hypertension only resolves after delivery of the placenta; therefore, it is widely accepted that abnormal placentation and/or placental dysfunction plays a causal role in the pathogenesis of PE [19,20].

The BPH/5 model is a spontaneous genetic model of early onset PE that shows mild hypertension and obesity, but without signs of renal disease, prior to pregnancy. The BPH/5 female mouse is unique in that, with pregnancy, she exhibits the diagnostic features of PE without pharmacological or surgical intervention nor transgenesis or targeted mutagenesis. Thus, with BPH/5 we can examine pre-pregnancy maternal health status as well as events during, and after pregnancy that may contribute to PE in women. Importantly, obesity-induced hypertension and hyperleptinemia, which are associated with increased inflammation, endothelial

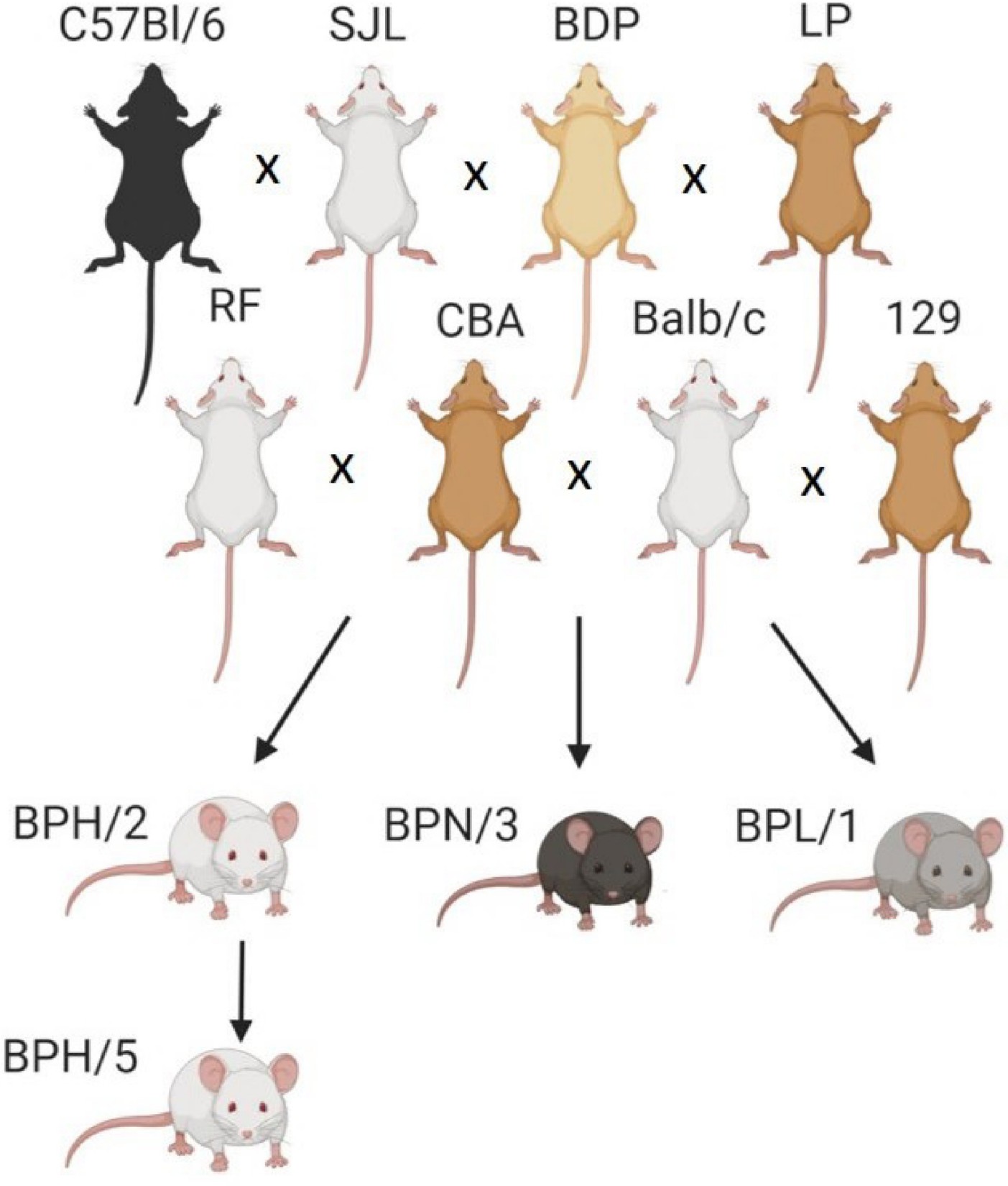

**Fig 1. Generation of the Blood Pressure High (BPH)/5 mouse.** BPH/5 mice were produced after more than twenty generations of brother-sister crosses of BPH/2 mice, which were derived alongside blood pressure normal (BPN)/3 and blood pressure low (BPL)/1 mice from an 8-way cross between C57Bl/6, Swiss Jim Lambert (SJL), BDP, LP, RF, CBA, BALB/c, and 129 inbred mice. Image made using BioRender.

dysfunction, and activation of the renin-angiotensin system (RAS), are observed in human PE [21–23] as well as in BPH/5 pregnancies [24–26]. Deciphering the genetic signature of mice predisposed to PE may help in identifying women who are at a heightened risk for developing PE in pregnancy.

Herein, we use a whole genome sequencing (WGS) approach to identify genetic defects in female BPH/5 mice that may be responsible for the cardiometabolic phenotypes driving their PE-like syndrome in pregnancy. We hypothesized that female BPH/2 and BPH/5 mice would be more genetically similar to each other than C57, the control mouse used in all published BPH/5 studies. We identified chromosomal variants in BPH/5 females that may contribute to genetic mutations and alterations in the proteome. The chromosomal variants are associated with immune system processes, which is congruent with previously described pathologies in female BPH/5 mice.

## Methods

All animals were maintained according to Louisiana State University Institutional Animal Care and Use Committee (IACUC) approved standards. Adult female (8-12-week-old) BPH/5 mice from an in-house colony (Louisiana State University), and BPH/2 and C57Bl/6 female mice purchased from Jackson Laboratory were used. All animals were maintained according to Louisiana State University Institutional Animal Care and Use Committee (IACUC) approved standards. Total Genomic DNA was isolated from tail snips of 3 mice belonging to each strain (BPH/5, BPH/2 and C57Bl/6) using DNeasy Blood & Tissue Kit from Qiagen, and sequencing libraries were generated using the Nextera DNA Flex library kit (Illumina, San Diego, CA). All samples were sequenced with paired end 150bp reads on the Illumina NovaSeq (Illumina, San Diego, USA). Raw reads were mapped to the mm10 reference genome using the Illumina DRAGEN Germline Pipeline (v3.2.8). Each mouse replicate variants were merged with bcftools (v1.10) and unique/common variants were pulled for cross comparisons with vcftools (v0.1.17). Variants were annotated for putative impacts with Ensembl Variant Effect Predictor (v101.0) to the mm10 genome. Gene ontology (GO) enrichment was performed via PANTHER to identify significantly enriched biological processes (FDR $p < 0.05$) and those ontologies present on chromosome 8 and X.

## Results

### BPH/5 and BPH/2 female mice share chromosomal variants, but are not genetically identical

Blood pressure in female BPH/5 mice is exacerbated by pregnancy [9]. Thus, with new onset proteinuria by mid- to late-gestation and severe hypertension, BPH/5 can be considered a model of superimposed PE [37]. The phenotypes of the non-pregnant female BPH/5 female mouse, including mild hypertension and obesity, are similar to those of women at higher risk for developing PE [10,13,27,28]. Therefore, determining the genetic underpinnings of these phenotypes can enhance our ability to identify women who are at high-risk before pregnancy. We performed WGS to identify single nucleotide polymorphisms (SNPs) and insertion/deletion variants as compared to the reference mm10 genome in female BPH/5, BPH/2 and C57 female mice, and present these as Venn Diagrams to illustrate the cross-sample comparisons

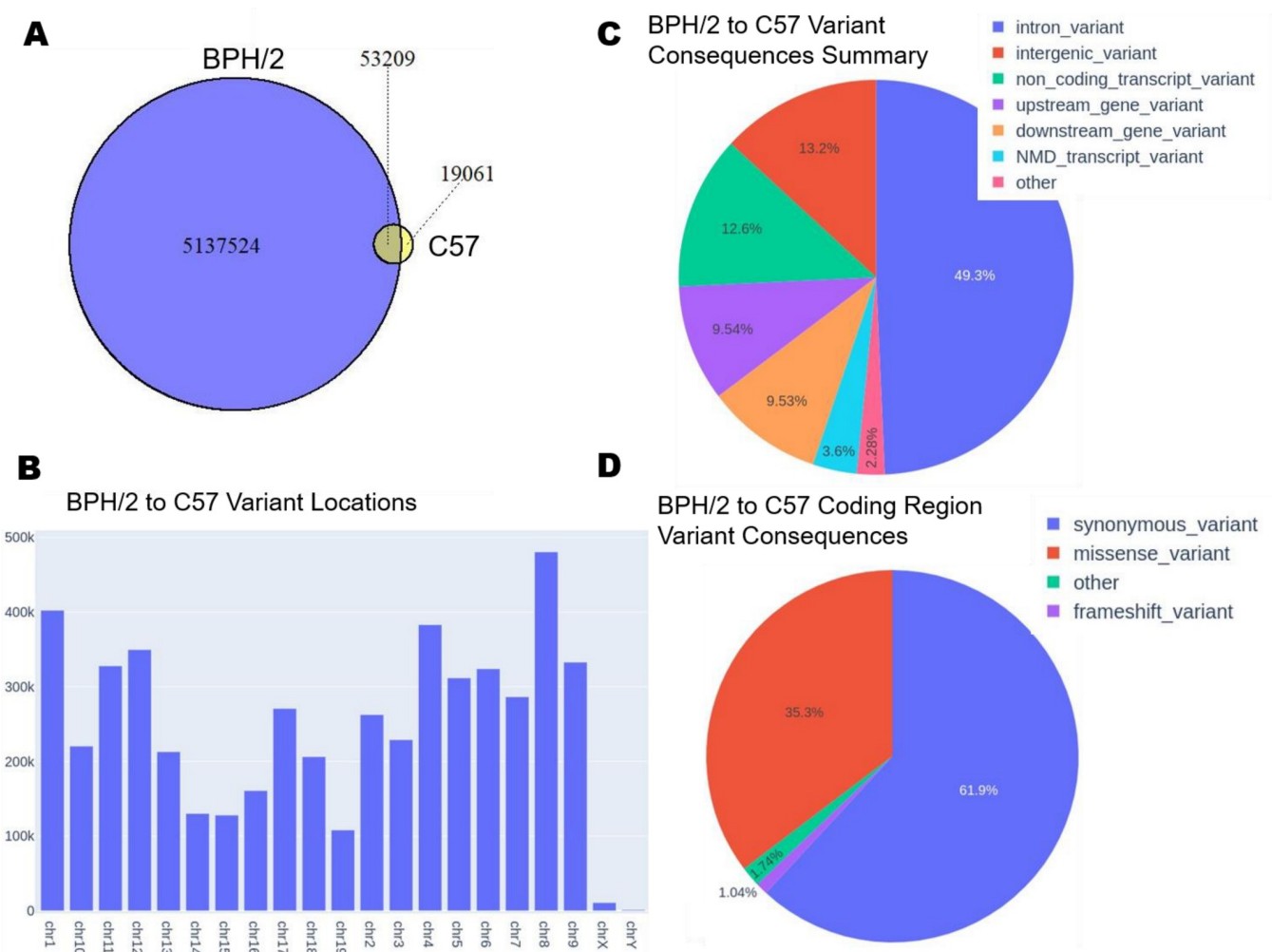

**Fig 2. Genetic comparison of Blood Pressure High/(BPH) 2 and C57 female mouse. A)** Venn diagram representing the quantified overlap of chromosomal variants between BPH/2 and C57. **B)** Location of variants along the chromosome between BPH/2 and C57. **C)** Pie chart representing the percentage of the types of putative variant consequences between BPH/2 and C57: Intron, intergenic, non-coding transcript, upstream of a gene, of a gene, nonsense mediated decay (NMD) transcript, and other (all variant types <1%). **D)** Pie chart representing the percentage of types of putative coding region variant consequences between BPH/2 and C57: Synonymous, missense, frameshift, and other (all variant types <1%).

(**Figs 2A, 3A and 4A**). There are 53,209 common chromosomal variants between BPH/2 and C57 with 5,137,524 unique to BPH/2 and 19,061 unique to C57 (**Fig 2A**). We found 4,334,355 common chromosomal variants between BPH/5 and BPH/2 with 856,378 unique to BPH/2 and 1,094,415 unique to BPH/5 (**Fig 3A**). Comparison of BPH/5 and C57 revealed 54,920 shared chromosomal variants, 5,373,850 unique to BPH/5, and 17,350 unique to C57 (**Fig 4A**). Therefore, while female BPH/5 and BPH/2 are genetically more similar to each other than to female C57 mice in terms of numbers of mutations, there are also distinct mutations present unique to each strain.

## The majority of BPH/5 and BPH/2 chromosomal variants in the coding region are missense mutations

Female BPH/5 mice exhibit a number of phenotypic differences when compared to normotensive female C57 mice [12]. While studies specifically comparing female BPH/5 and BPH/2

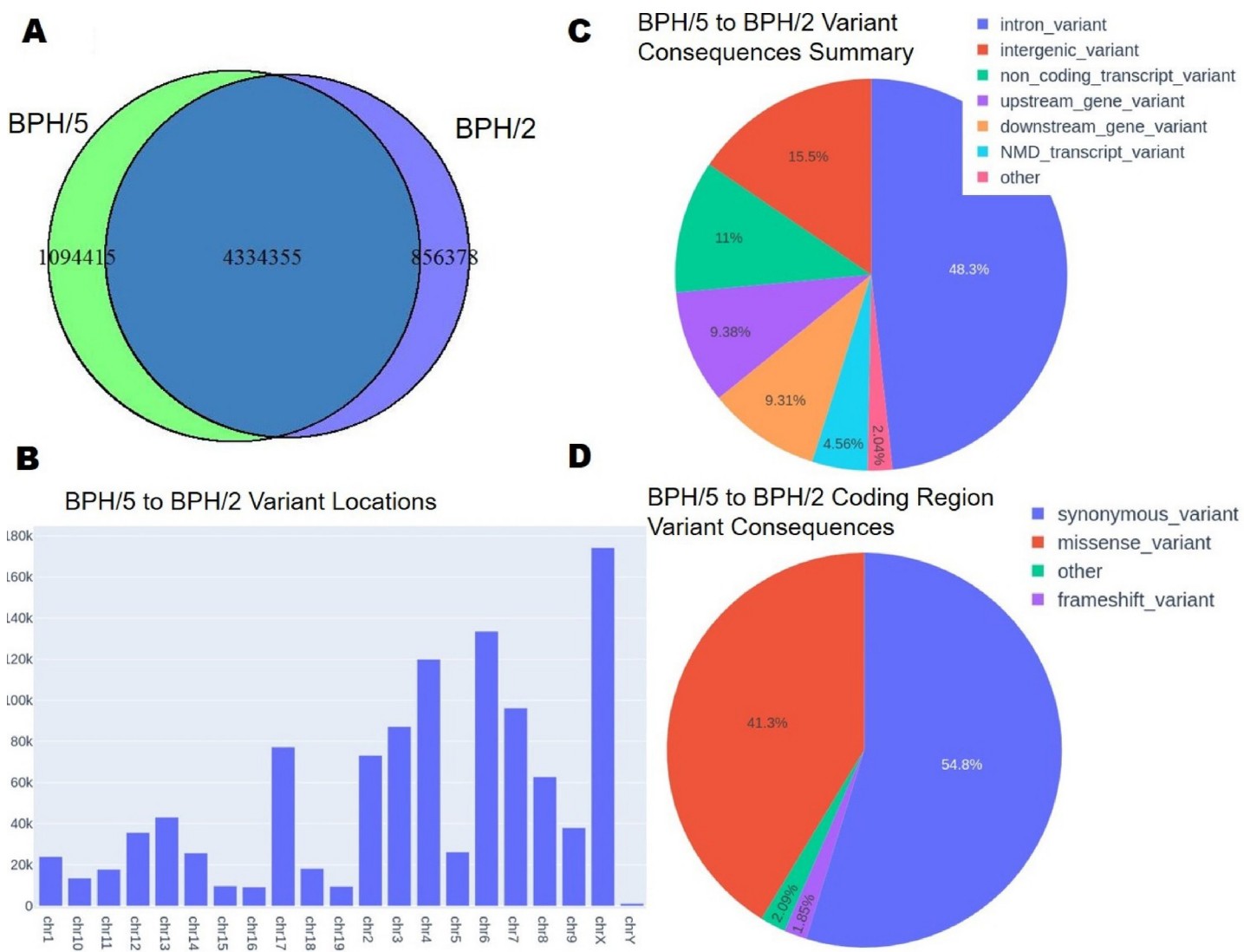

**Fig 3. Genetic comparison of Blood Pressure High (BPH)/5 and 2 female mouse. A)** Venn diagram representing the quantified overlap of chromosomal variants between BPH/5 and BPH/2. **B)** Location of variants along the chromosome between BPH/5 and BPH/2. **C)** Pie chart representing the percentage of putative variant consequences between BPH/5 and BPH/2: Intron, intergenic, non-coding transcript, upstream of a gene, downstream of a gene, nonsense mediated decay (NMD) transcript, and other (all variant types <1%). **D)** Pie chart representing the percentage of types of putative coding region variant consequences between BPH/5 and BPH/ 2: Synonymous, missense, frameshift, and other (all variant types <1%).

mice have not yet been performed, we hypothesized that their genetic makeup would be similar due to their shared parentage. We determined the number of variants per chromosome. Comparison of mutations between BPH/2 and C57 females demonstrated that chromosome 8 has the largest number of unique variants (**Fig 2B**). The majority of these are intronic (49.3%), while 13.2% are intergenic (**Fig 2C**). Furthermore, of the non-synonymous consequences, 35.3% are attributed to missense mutations (**Fig 2D**). Fewer unique chromosomal variants were identified in the comparison of BPH/5 versus BPH/2, with the largest number located on chromosome X (**Fig 3B**). As in the BPH/2 versus C57 comparison, most variants are intronic (48.3%), while 15.5% are intergenic (**Fig 3C**) and of the non-synonymous consequences, 41.3% are missense mutations (**Fig 3D**). As with BPH/2 versus C57, the largest number of unique variants identified in the comparison of BPH/5 and C57 are on chromosome 8

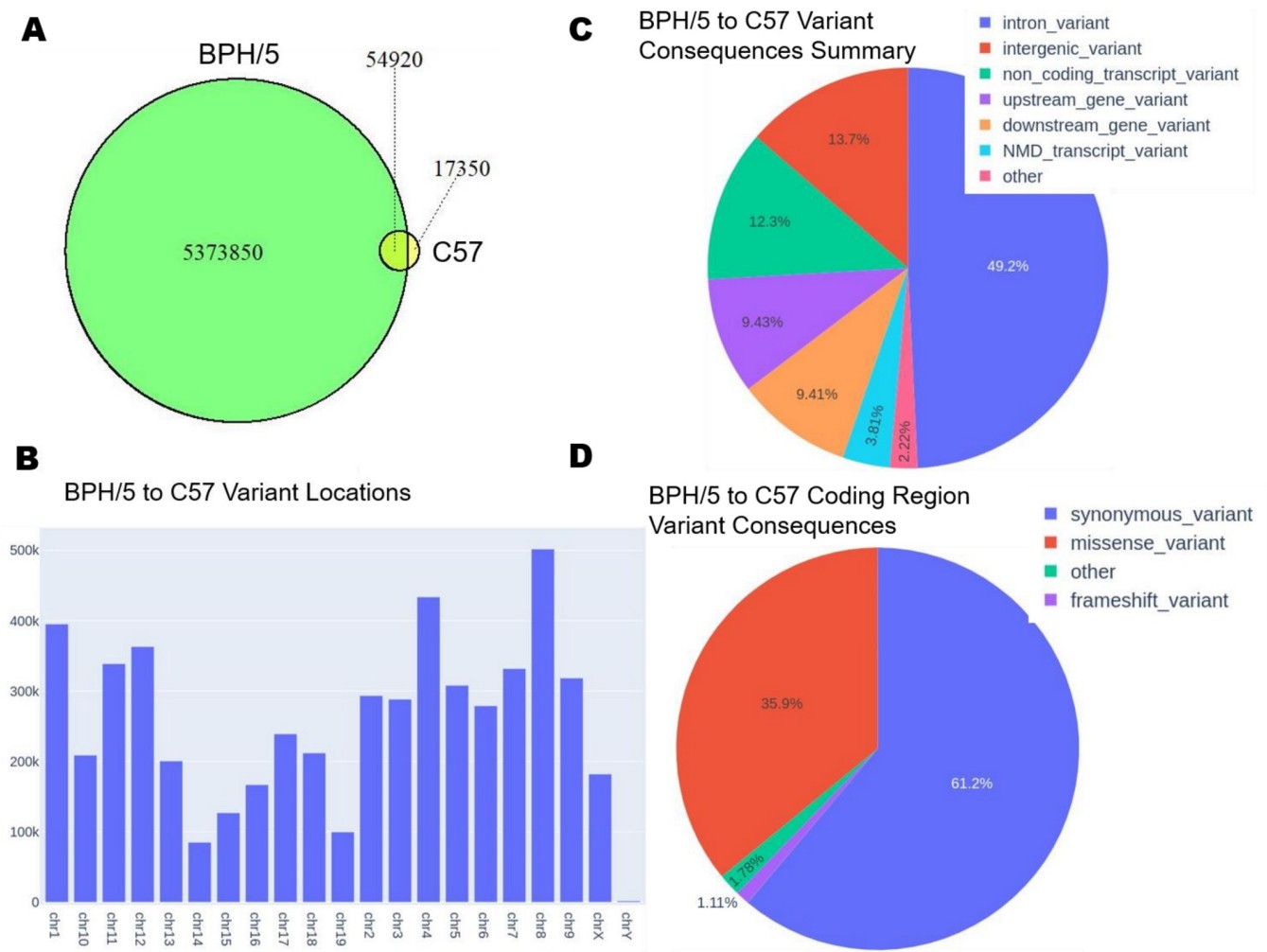

**Fig 4. Genetic comparison of Blood Pressure High (BPH)/5 and C57 female mouse. A)** Venn diagram representing the quantified overlap of chromosomal variants between BPH/2 and C57. **B)** Location of variants along the chromosome between BPH/2 and C57. **C)** Pie chart representing the percentage of the types of putative variant consequences between BPH/2 and C57: Intron, intergenic, non-coding transcript, upstream of a gene, nonsense mediated decay (NMD) transcript, and other (all variant types <1%). **D)** Pie chart representing the percentage of types of putative coding region variant consequences between BPH/2 and C57: Synonymous, missense, frameshift, and other (all variant types <1%).

(**Fig 4B**). The majority of these are intronic (49.2%), while 13.7% are intergenic (**Fig 4C**). Of the non-synonymous consequences, 35.9% are missense mutations (**Fig 4D**). Because SNPs producing missense mutations can result in amino acid substitutions, functional consequences of such variants are likely to contribute to the BPH/5 phenotypes.

## Predicted altered biological processes in BPH/5 and BPH/2 are predominately immune-related

To better understand the importance of the chromosomal variation and mutational burden of the BPH/5 model as compared to C57 and to the immediate strain they were derived from, BPH/2, GO enrichment analysis was performed. The full GO list revealed differences in immune-related programs when comparing biological processes enriched in BPH/2 versus C57, BPH/5 versus BPH/2, and BPH/5 versus C57 (**S1**–**S3 Tables**). The significantly enriched biological processes associated with hypertensive phenotypes observed in BPH/2 versus C57

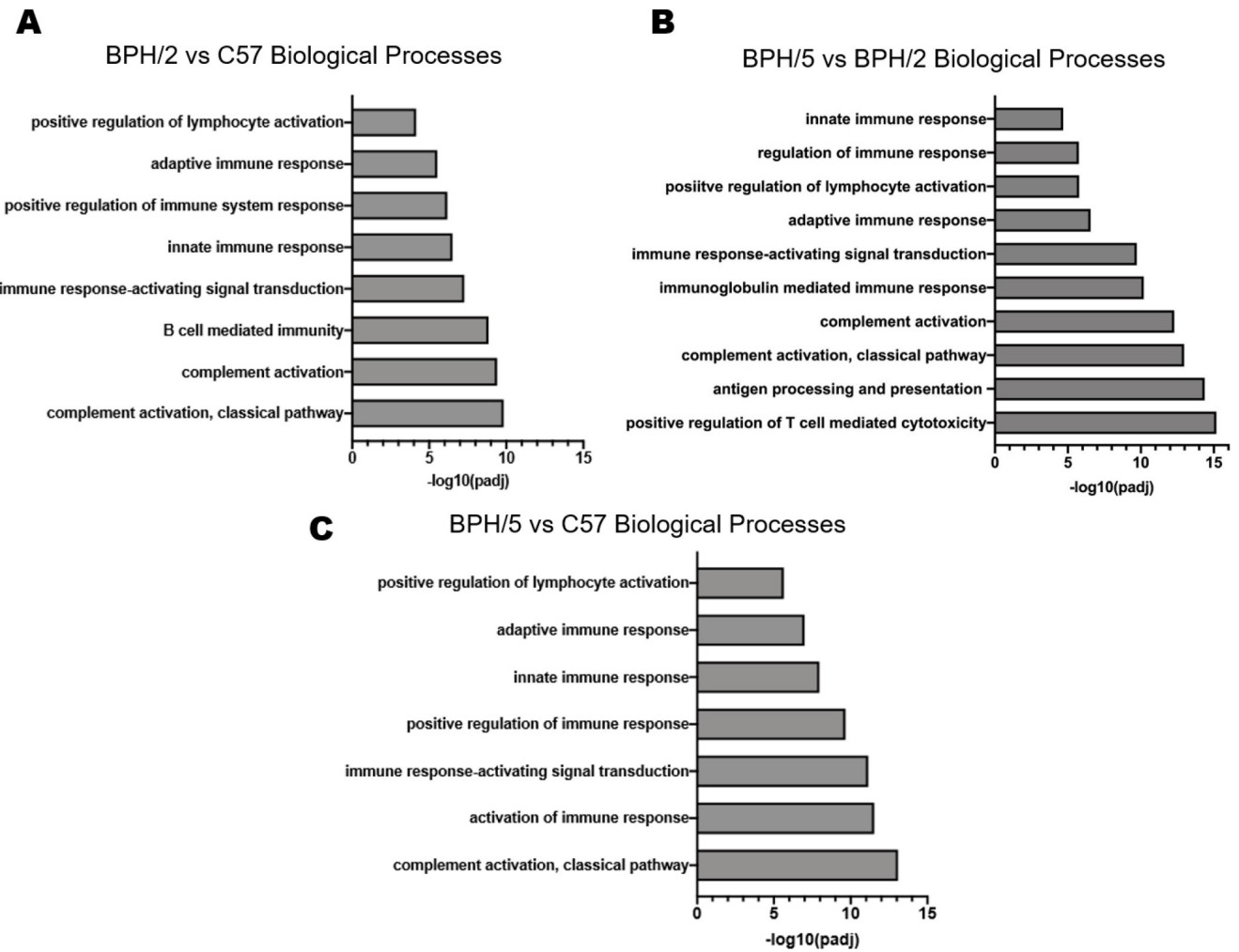

**Fig 5. Gene ontology of relevant biological processes perturbed in hypertensive mouse strains, Blood Pressure High (BPH)/5 and 2 female mouse. A)** Top gene ontology (GO) terms of biological processes altered in BPH/2 versus C57 ranked by significance **B)** Top GO terms of biological processes altered in BPH/5 versus BPH/2 ranked by significance **C)** Top GO terms of biological processes altered in BPH/5 versus C57 ranked by significance.

(**Fig 5A**), BPH/5 versus BPH/2 (**Fig 5B**), and BPH/5 versus C57 (**Fig 5C**) female mice include those involved in adaptive and innate immunity; notably, complement activation and lymphocyte regulation. Genetic variants are present between BPH/5 and BPH/2 on chromosome X and map to 69 genes (**S4 Table**). Although not significant, biological processes implicated include cellular and developmental processes, and reproduction (**Fig 6A**). When comparing female BPH/5 to C57, 336 genetic mutations were identified on Chromosome 8 (**S5 Table**) and a significant number of biological processes were altered, including cellular, metabolic, developmental, and reproductive (**Fig 6B**). Raw sequencing data have been uploaded to the Short Reads Archive with project number PRJNA701765 Processed annotated data has been provided as S6 and S7 Tables for BPH/5 versus BPH/2 (**S6 Table**) and BPH/5 versus C57 (**S7 Table**; **accessible online:** 10.6084/m9.figshare.14888202).

## Discussion

PE is diagnosed in the latter half of pregnancy; however, the origins of early onset PE are thought to begin early in pregnancy at the time of placentation, or even before conception.

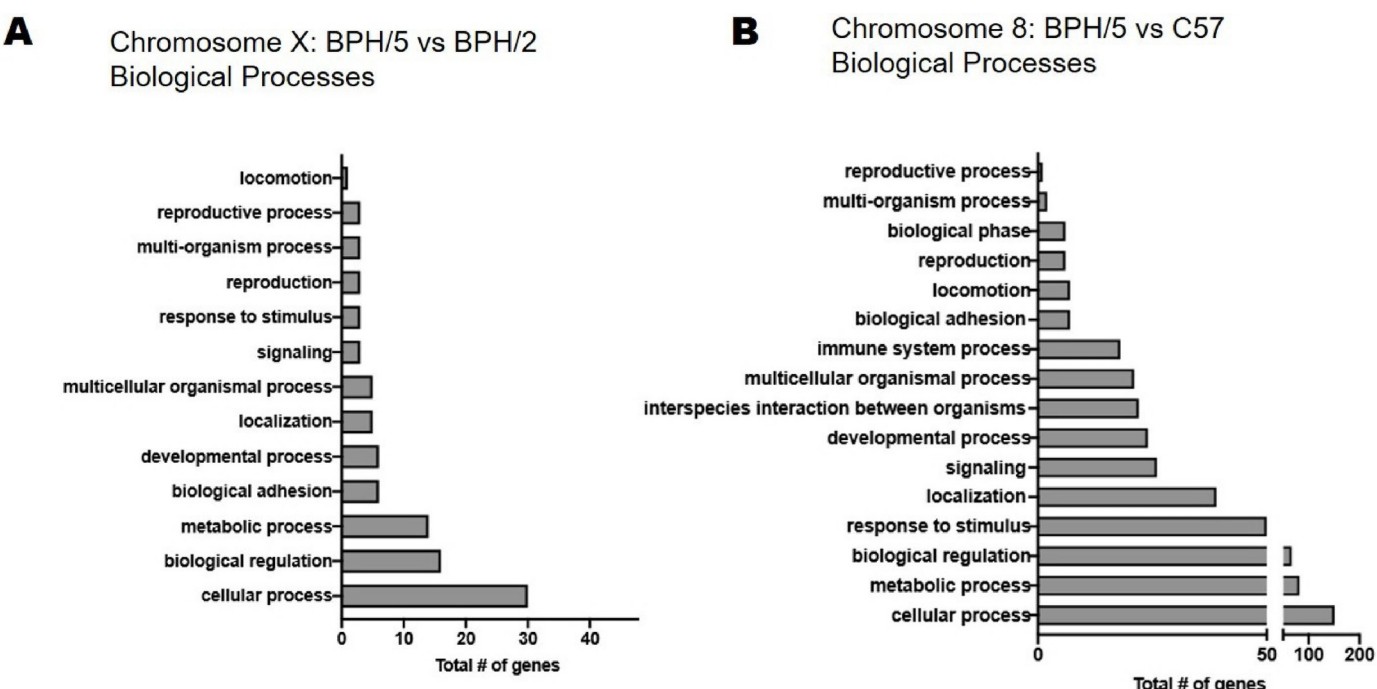

**Fig 6. Gene ontology of relevant biological processes perturbed in the Blood Pressure High (BPH)/5 female mouse. A)** Top gene ontology (GO) terms of biological processes altered in BPH/5 versus BPH/2 on Chromosome X and **B)** Top GO terms of biological processes altered BPH/5 versus C57 in ranked by significance.

Therefore, identifying high-risk women would allow for earlier detection, increased prenatal medical management, and potentially improved maternal and fetal outcomes. The mouse model of superimposed PE, BPH/5, has been utilized to study the pathogenesis of PE for several decades. While the maternal and fetal PE phenotypes in BPH/5 are well described, their genotypic determinants are unknown. We utilized Illumina short-read sequencing technology to perform WGS to identify genetic differences (chromosomal variation and mutational burden) followed by gene ontology to predict perturbed programs in female BPH/5 versus female BPH/2 and C57, which has been the control strain for all BPH/5 published studies [10]. We found that female BPH/5 are similar to female BPH/2 mice, while also presenting unique gene variants not observed in BPH/2. Furthermore, the differences between female BPH/5 and normotensive C57 mice are due primarily to missense mutations within genes involved in immune-related processes.

There are few reports on the BP mice, and even fewer in which the strains are compared to one another. Our findings suggest sex specific differences in the cardiometabolic profiles of the closely related BP strains, highlighting the importance of acknowledging sex as a biological variable in genetic models of disease. Although the metabolic profile of adult female BPH/2 mice has not been fully elucidated, BPH/2 females demonstrate increased blood pressure and heart rates as well as endothelial dysfunction and decreased body weight compared to BPN/3 females [3,7]. The maternal and fetal characteristics of BPH/2 pregnancies have not yet been reported. Thus, it is unknown whether female BPH/2 mice could be a mouse model of superimposed PE, similar to BPH/5. Male BPH/2 mice have been utilized to investigate basic cardiometabolic physiology. In adulthood, they are hypertensive with a greater metabolic rate and decreased body weight compared to normotensive BPN/3 males [2], which is similar to what is observed in comparison of BPH/2 and BPN/3 females. Genome Wide Association Studies

(GWAS) of male BPH/2 mice has been undertaken to elucidate the genetic defects associated with hypertension in this strain [4], but no such studies have been performed on BPH/5 male or female mice.

BPH/5 females have pre-existing cardiometabolic risk, including hyperphagia, obesity and leptin resistance, and hypoestrogenemia [12], before pregnancy and their phenotype is unique from BPH/5 males. Interestingly, BPH/5 males are not obese, nor do they share all the cardio-metabolic phenotypes observed in their female littermates [29]. Female offspring inherit two X chromosomes and inactivation of one is necessary to prevent overdose of X-linked genes. Several cardiometabolic diseases have been associated with X overdose and sex-biased gene expression [30–35]. This phenomenon may exist in the BPH/5 model and further investigation is needed. For example, mutations in the Fanconi anemia group B (FANCB) gene are present in men with X-linked Fanconi anemia and are associated with infertility in male mice [36,37]. BPH/5 female mice have genetic mutations in *Fancb*, which may undergo X-inactivation as they are not anemic nor infertile (unpublished data).

Comparison of normotensive C57 females to hypertensive BPH/2 and to borderline hypertensive BPH/5 females, showed that the largest number of unique chromosomal variants are on chromosome 8 for both BPH/2 versus C57 and BPH/5 versus C57. In a hypertensive rat model, a quantitative trait locus for elevated heart rates was found on chromosome 8 [38]. RAS genes were considered as they are important for cardiovascular adaptations in mammals, including blood pressure regulation, and may contribute to female BPH/5 borderline hypertension before pregnancy. We hypothesized that BPH/5 and BPH/2 mice would possess similar mutations in RAS genes, specifically angiotensinogen (Agt) and kallikrein, as previously described in BPH/2 [2]. BPH/5 female mice have non-synonymous mutations in Agt, but not in angiotensinogen converting enzyme (Ace) and Agt receptor (Agtr) 1a. Importantly, Agt is located on Chromosome 8 in the mouse and is critically important for downstream RAS function; therefore, the Agt mutation in BPH/5 may be enough to promote the cardiovascular defects (tachycardia and mild hypertension) that are exacerbated by pregnancy in this model. Interestingly, kallikrein gene mutations are not observed in BPH/5 female mice, which supports our findings here that BPH/5 are similar to BPH/2 but uniquely different. Research is ongoing in our laboratory to fully describe all RAS genes in the BPH/5 genome. Several PE studies implicate RAS polymorphisms in the maternal genome, such as ACE and aldosterone synthase, as associated with adverse outcomes [39–41]. However, there is significant variation when looking at women from different geographic backgrounds [39,42]. Thus, a large consortium of data is needed from women of diverse ethnic backgrounds to adequately assess the polymorphic contributors of PE-related traits and the impact of maternal genetics on PE [43].

Mutations on chromosome X account for the majority of variants present between female BPH/5 and BPH/2 female mice. Although GO analysis did not identify significantly different biological processes between the two strains, mutations were present in genes involved in immune system function, including growth factor receptor bound protein 2-associated protein 3 (Gab3) in BPH/5 strain. Proteins in the Gab family are intracellular scaffolding and docking molecules sensitive to growth factors, cytokines, and antigen receptors. They are important for immune cell function, including interleukin (IL) 15 induced natural killer (NK) cell expansion [44] and macrophage differentiation [45]. While non-pregnant Gab3 deficient mice do not show defects in development, hematopoiesis, and immune cell function, they have significant abnormalities of pregnancy which are shared by pregnant BPH/5 mice and preeclamptic women [46]. Gab3 deficient mice have abnormal placentation due to impaired uterine NK cell priming and expansion by IL-2 and IL-15 resulting in maternal and fetal morbidity and mortality [44]. BPH/5 female mice spontaneously develop a PE-like phenotype in pregnancy, which include abnormalities in placental development such as shallow trophoblast invasion

and inadequate spiral artery remodeling [24]. This is thought to contribute to hypoxia and inflammation at the maternal-fetal interface, which have been shown to be causally related to maternal hypertension and fetal growth restriction in this model [47,48]. BPH/5 females also have fewer uterine NK cells compared to C57 in pregnancy and restoration of uterine NK cells was associated with improved pregnancy outcomes [48]. Further investigations are warranted to better understand the role that Gab3 mutations might play in BPH/5 pregnancy outcomes.

In addition to uterine NK cell defects, BPH/5 female mice also have inflammatory immune cell activation at the maternal-fetal interface, early in pregnancy [49]. Immunotolerance at the maternal-fetal interface is key for embryo survival and pregnancy success [50]. An aberrant maternal immune response and subsequent lack of immunotolerance to the early embryo has been proposed as a central mechanism in the pathogenesis of PE [51]. In the post-implantation, pre-placentation decidua, BPH/5 have decreased macrophage, but increased T lymphocyte populations (CD3+, CD69+, CD4+, CD8+) along with decreased anti-inflammatory IL-10 [49]. Data from Gab3 deficient mice suggest that pre-conception dysregulation of immune cells may not be apparent until challenged with pregnancy. This supports the hypothesis that genetic mutations prior to pregnancy may be useful biomarkers for development of PE in high-risk women.

In line with our pregnancy studies in BPH/5, GO enrichment analyses showed that immune system processes account for some of the genetic differences between BPH/5 and C57 female mice. Broadly, innate and adaptive immune responses are perturbed in BPH/5, including complement activation and T cell function [49,52,53]. Aberrations in T and B lymphocyte function as well as the complement system are considered pro-hypertensive [54]. BPH/5 have increased complement factor 3 (C3) in adipose tissue before and during pregnancy as well as at the maternal-fetal interface that is attenuated by caloric restriction via pair-feeding beginning at conception [55,56]. Thus, genetic defects in immune system processes may contribute to the abnormal cardiometabolic profile, including hypertension and obesity, observed in BPH/5 females. Taken together, this suggests that dietary intervention and reversal of obesity could influence the genetic propensity for immune cell dysregulation in BPH/5 female mice.

Female BPH/5 mice have tachycardia and increased blood pressure along with obesity and dyslipidemia prior to pregnancy [12]. Women with pre-existing hypertension have a 25% increased risk for developing PE during pregnancy [57]. When this occurs, it is diagnosed as superimposed PE in contrast to new onset PE. Other pre-conception maternal risk factors include being overweight or obese with a body mass index (BMI) between 25–30 kg/m$^2$ and greater than 30 kg/m$^2$, respectively [58]. There is a strong correlation between BMI and adverse pregnancy outcomes with obese women have a 3-times increased risk for developing PE compared to lean counterparts [58]. This is further supported by data showing weight loss before pregnancy via bariatric surgery decreased PE risk in obese pregnant women [59,60]. Interestingly, while pre-existing obesity and excessive gestational weight gain is associated with increased PE risk, only 10% of obese women develop PE [28,61]. This may be due to differences in circulating lipids with a strong correlation of PE and increased triglycerides and free fatty acids [28]. Understanding the differences between obese females, that go on to have PE versus the ones that do not, is important. The BPH/5 model affords the opportunity to investigate cardiometabolic risk before pregnancy and how it contributes to development of a PE-like phenotype. Several hypotheses exist to explain the association between obesity and PE. For example, reduced nitric oxide availability secondary to increased asymmetrical dimethyl arginine (ADMA) and oxidative stress, increased sympathetic tone, and increased expression of angiotensinogen from adipose tissue have been proposed. Simply, increased systemic inflammation as a result of circulating adipokines, such as leptin, have been implicated in the

onset of PE. Mechanistic studies are necessary to understand the role of adipose tissue, ADMA, and inflammation as well as oxidative stress in PE.

Given what is known regarding obesity and its implications on the cardiovascular system, it is not surprising that BPH/5 female exhibit mild hypertension even before pregnancy. Recent research suggests that the relationship between maternal obesity and exacerbation of cardiovascular disease in pregnancy is modulated by the maternal microbiome [62]. In humans, obese mothers have a dysbiotic gut microbiome characterized by a high firmicutes-to-bacteroidetes ratio [8]. This increased ratio has been linked to hypertension in humans and in animal models. In the context of PE, maternal gut dysbiosis may lead to an increased inflammatory response which could contribute to the development of PE. Chen et al characterized an increase of Proteobacteria, Bacteroidetes, and Enterobacteriaceae in women that developed PE [63]. It has also been proposed that bacterial translocation to the placenta may occur in pregnancy provoking an inflammatory response promoting abnormal placentation, and predisposing offspring to gut microbial dysbiosis as well [9]. Although this idea is controversial, microbial populations may be involved in the aberrant immune system activation and inflammation seen in PE. The BPH/5 female microbiome is currently being characterized using metagenomic approaches.

BPH/5 are an inbred genetic model of superimposed PE that spontaneously develops maternal and fetal features of PE by late gestation. Although this model has been utilized to investigate events before and during pregnancy that may contribute to PE in women, the genetic origins of PE in BPH/5 were previously unexplored. Herein, we characterize the genetic underpinnings of the PE phenotype in BPH/5 female mice. We found not one, but many distinctive genetic variations in BPH/5 female mice, even when compared to the closely related BPH/2 strain. Largescale genomic studies are necessary to further explore the genetics of different rodent strains utilized to study PE. These findings may provide new avenues of PE research, including identification of genetic biomarkers of PE in women. If genetic determinants of the BPH/5 phenotype are implicated in human PE, these mice will be particularly useful for *in vivo* pregnancy studies to explore prevention, treatment, and long-term prognosis of women and offspring affected by PE.

## Supporting information

**S1 Table.**
(XLS)

**S2 Table.**
(XLS)

**S3 Table.**
(XLS)

**S4 Table.**
(XLSX)

**S5 Table.**
(XLSX)

**S6 Table.**
(ZIP)

**S7 Table.**
(DOCX)

## Acknowledgments

Dr. Robin Davisson for the generous gift of the BPH/5 mice. Drs. Carrie J. Shawber, Andrea Johnston, and Anik Boudreau, Chris Morrison for manuscript review.

## Author Contributions

**Conceptualization:** Jenny L. Sones, Christina C. Yarborough, Nataki C. Douglas.

**Data curation:** Jenny L. Sones, Christina C. Yarborough, Valerie O'Besso, Alexander Lemenze, Nataki C. Douglas.

**Formal analysis:** Jenny L. Sones, Alexander Lemenze, Nataki C. Douglas.

**Investigation:** Jenny L. Sones.

**Software:** Alexander Lemenze.

**Supervision:** Nataki C. Douglas.

**Validation:** Alexander Lemenze, Nataki C. Douglas.

**Writing – original draft:** Jenny L. Sones.

**Writing – review & editing:** Jenny L. Sones, Christina C. Yarborough, Alexander Lemenze, Nataki C. Douglas.

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
