## [Decision Letter · Decision Letter 0]

21 Apr 2021

PONE-D-21-04771

Genotypic analysis of the female BPH/5 mouse, a model of superimposed preeclampsia

PLOS ONE

Dear Dr. Sones,

Thank you for submitting your manuscript to PLOS ONE. After careful consideration, we feel that it has merit but does not fully meet PLOS ONE’s publication criteria as it currently stands. Therefore, we invite you to submit a revised version of the manuscript that addresses the points raised during the review process.

We look forward to receiving your revised manuscript.

Kind regards,

Michael Bader

Academic Editor

PLOS ONE

Journal Requirements:

2. As part of your revisions, please indicate if you used anesthesia for the tail snip procedure and if so, please specify. Thank you for your attention to this matter.

Reviewers' comments:

Reviewer's Responses to Questions

**Comments to the Author**

1. Is the manuscript technically sound, and do the data support the conclusions?

Reviewer #1: Yes

Reviewer #2: Yes

2. Has the statistical analysis been performed appropriately and rigorously? 

Reviewer #1: I Don't Know

Reviewer #2: I Don't Know

3. Have the authors made all data underlying the findings in their manuscript fully available?

Reviewer #1: Yes

Reviewer #2: Yes

4. Is the manuscript presented in an intelligible fashion and written in standard English?

Reviewer #1: Yes

Reviewer #2: Yes

5. Review Comments to the Author

Reviewer #1: I believe these findings are very important to publish and also make the dataset available to other researchers. To be able to identify gene mutations that may predispose these mice to spontaneous preeclampsia and correlate with mutations found in human preeclamptic patients would be a very useful research tool.

I have a few relatively minor questions/comments.

I assume the BPH/2 strain is included as a genetically related but non-preeclamptic hypertensive strain, is that correct? Is there evidence from yourself or from other publications that the BPH/2J mice are not also preeclamptic? If it is unknown whether the BPH/2 are preeclamptic, then how do you know which gene mutations are associated with preeclampsia as opposed to hypertension alone. Differences may also be associated with being lean (BPH/2) versus obese (BPH/5). Can you elaborate more in the discussion?

Is 3 animals per strain an adequate sample size for whole genome sequence analysis?

Can you please expand on how the BPH/5 diverged from the BPH/2 mice strain in the original breeding protocol? Were they crossed with a strain other than the BPH/2?

The writing on the figures 2-4 is mostly too small to read.

In Supp table 4 and 5 there is no header for the columns? Are these just mutations in protein coding regions? A descriptive title would be useful so the reader doesn’t have to go back to the text to figure out what strains are being compared and column heading please. I think the value of this study lies in the ability of other researchers to probe this dataset to look for genes of interest both now and as new findings become available. This will be a really useful tool for other researchers for years to come. Are the BPH/5 mice available to other researchers to study, either from yourself or from somewhere like Jackson laboratories? Is it possible or feasible for the author to detail mutations in non-protein coding regions similar to Supp table 4 and 5, but those that may occur in known regulatory regions such as promotor or enhancer regions for a gene?

Reviewer #2: In this study, Sones et al use a whole genome approach to identify genetic defects in the BPH/5 mouse model of preeclampsia, that may explain the predisposition to risk of PE in women. I have some comments/suggestions below:

-The abstract would be strengthened by mentioning some of the other differences, and being more specific.

-The relevance of the BPH/2 mouse is not clear. Although this is explained within the paper, I would suggest explaining the relationship/phenotype of the BPH/2.

-Can any further information be derived regarding changes in specific genes related to the immune responses altered? --Can any specific genes of interest be identified?

-The susceptibility to PE in obese women is discussed in the introduction. The authors suggest that the findings from this study may be useful in understanding the relationship between obesity and PE, but it is not addressed in the discussion. Even if nothing specific was identified, this could be tied into immune response?

-Generally speaking, the discussion needs to be better linked to the hypothesis stated in the introduction.

Many of the figures were not large enough for me to read

6. PLOS authors have the option to publish the peer review history of their article (what does this mean?). If published, this will include your full peer review and any attached files.

Reviewer #1: No

Reviewer #2: No

---

## [Author Response · Author response to Decision Letter 0]

1 Jun 2021

Reviewer #1: I believe these findings are very important to publish and also make the dataset available to other researchers. To be able to identify gene mutations that may predispose these mice to spontaneous preeclampsia and correlate with mutations found in human preeclamptic patients would be a very useful research tool.

I have a few relatively minor questions/comments.

I assume the BPH/2 strain is included as a genetically related but non-preeclamptic hypertensive strain, is that correct? Is there evidence from yourself or from other publications that the BPH/2J mice are not also preeclamptic? If it is unknown whether the BPH/2 are preeclamptic, then how do you know which gene mutations are associated with preeclampsia as opposed to hypertension alone. Differences may also be associated with being lean (BPH/2) versus obese (BPH/5). 

The authors appreciate you highlighting this important point. To date, we cannot find a publication supporting BPH/2 females developing a preeclamptic phenotype. Therefore, the genetic differences between the 2 strains may only contribute to the BPH/5 obesity phenotype with BPH/2 females being lean compared to BPN/3 and BPH/5. However, in support of BPH/5 as a model of PE, BPH/5 females have new onset endothelial dysfunction with pregnancy while BPH/2 females have preexisting endothelial dysfunction. Thus, further investigations beyond the scope of this paper are needed in BPH/2 females to ascertain if they develop a PE-like phenotype as well.. These points have been added to the Introduction and Discussion.

Is 3 animals per strain an adequate sample size for whole genome sequence analysis? 

Yes, we were able to detect significant differences between strains with n=3. 

Can you please expand on how the BPH/5 diverged from the BPH/2 mice strain in the original breeding protocol? Were they crossed with a strain other than the BPH/2?

To our knowledge, only BPH/2 brothers and sisters were used to generate the BPH/5 subline as described by Schlager and is referenced in the Introduction and illustrated in Figure 1. 

The writing on the figures 2-4 is mostly too small to read. 

Thank you for pointing that out. We have enlarged the figures and the text within including color legends. 

In Supp table 4 and 5 there is no header for the columns? Are these just mutations in protein coding regions? 

Headings have been added. 

I think the value of this study lies in the ability of other researchers to probe this dataset to look for genes of interest both now and as new findings become available. This will be a really useful tool for other researchers for years to come. Are the BPH/5 mice available to other researchers to study, either from yourself or from somewhere like Jackson laboratories? 

Once the genetics of this model have been described and published, we plan to share the model broadly with the caveat that, as seen with this study, the genetic mutations leading to this phenotype are complex. 

 Is it possible or feasible for the author to detail mutations in non-protein coding regions similar to Supp table 4 and 5, but those that may occur in known regulatory regions such as promotor or enhancer regions for a gene? 

Full listing of mutations between BPH/5 v BPH/2 an BPH/5 v C57 have been included now as Supplemental table 6 and 7 respectively. 

Reviewer #2: In this study, Sones et al use a whole genome approach to identify genetic defects in the BPH/5 mouse model of preeclampsia, that may explain the predisposition to risk of PE in women. I have some comments/suggestions below:

-The abstract would be strengthened by mentioning some of the other differences, and being more specific. 

Thank you. The abstract now includes specific examples of the type of genetic differences we unearthed between BPH/5, C57Bl/6 and BPH/2. 

-The relevance of the BPH/2 mouse is not clear. Although this is explained within the paper, I would suggest explaining the relationship/phenotype of the BPH/2. 

The authors thank you for highlighting this deficiency. Because the closest genetic relative of BPH/5 is BPH/2, this comparison was included to determine how far the strains diverged genetically in light of their differing phenotypes. This has been elaborated in the Discussion to clarify for both Reviewer 1 and 2. 

-Can any further information be derived regarding changes in specific genes related to the immune responses altered? --Can any specific genes of interest be identified? 

Thank you for your comment. Full listing of mutations between BPH/5 v BPH/2 an BPH/5 v C57 have been included now as Supplemental table 6 and 7 respectively. The authors agree it will be important to follow up this descriptive study by validating genetic mutations on the mRNA as well as protein level using functional validation studies. This is ongoing in the laboratory and the focus of the follow up manuscript in preparation. 

-The susceptibility to PE in obese women is discussed in the introduction. The authors suggest that the findings from this study may be useful in understanding the relationship between obesity and PE, but it is not addressed in the discussion. Even if nothing specific was identified, this could be tied into immune response? -Generally speaking, the discussion needs to be better linked to the hypothesis stated in the introduction. 

The Discussion has been revamped to include a better explanation of BP lines, male and female, as well as the relevance of BPH/5 to obesity in women tying in with the immune system relevance. 

Many of the figures were not large enough for me to read 

Thank you for pointing that out. We have enlarged the figures and the text within including color legends. ________________________________________

---

## [Editor Report · Decision Letter 1]

7 Jun 2021

Genotypic analysis of the female BPH/5 mouse, a model of superimposed preeclampsia

PONE-D-21-04771R1

Dear Dr. Sones,

We’re pleased to inform you that your manuscript has been judged scientifically suitable for publication and will be formally accepted for publication once it meets all outstanding technical requirements.

Kind regards,

Michael Bader

Academic Editor

PLOS ONE
---

## [Editor Report · Acceptance letter]

2 Jul 2021

PONE-D-21-04771R1 

Genotypic analysis of the female BPH/5 mouse, a model of superimposed preeclampsia 

Dear Dr. Sones:

I'm pleased to inform you that your manuscript has been deemed suitable for publication in PLOS ONE. Congratulations! Your manuscript is now with our production department. 

Kind regards, 

on behalf of

Prof. Michael Bader 

Academic Editor

PLOS ONE